# COVID-19 Infection in Autosomal Dominant Polycystic Kidney Disease and Chronic Kidney Disease Patients: Progression of Kidney Disease

**DOI:** 10.3390/biomedicines12061301

**Published:** 2024-06-12

**Authors:** Silvia Lai, Francesca Tinti, Adolfo Marco Perrotta, Luca Salomone, Rosario Cianci, Paolo Izzo, Sara Izzo, Luciano Izzo, Claudia De Intinis, Chiara Pellicano, Antonietta Gigante

**Affiliations:** 1Department of Translational and Precision Medicine, UOC Nephrology, Sapienza University of Rome, 00185 Rome, Italy; francesca.tinti@uniroma1.it (F.T.); adolfomarco.perrotta@uniroma1.it (A.M.P.); luca.salomone@uniroma1.it (L.S.); rosario.cianci@uniroma1.it (R.C.); 2Department of Surgery “Pietro Valdoni”, Policlinico Umberto I, Sapienza University of Rome, 00185 Rome, Italy; p_izzo@hotmail.it (P.I.); luciano.izzo@uniroma1.it (L.I.); deintinis.1891513@studenti.uniroma1.it (C.D.I.); 3Plastic Surgery Unit, Multidisciplinary Department of Medical-Surgical and Dental Specialties, University of Campania “Luigi Vanvitelli”, 80138 Naples, Italy; sa_izzo@hotmail.it; 4Department of Translational and Precision Medicine, Sapienza University of Rome, 00185 Rome, Italy; chiara.pellicano@gmail.com (C.P.); antonietta.gigante@uniroma1.it (A.G.)

**Keywords:** autosomal dominant polycystic kidney disease, COVID-19 infection, chronic kidney disease, SARS-CoV-2 virus

## Abstract

Introduction: the COVID-19 pandemic has brought to light the intricate interplay between viral infections and preexisting health conditions. In the field of kidney diseases, patients with Autosomal Dominant Polycystic Kidney Disease (ADPKD) and Chronic Kidney Disease (CKD) face unique challenges when exposed to the SARS-CoV-2 virus. This study aims to evaluate whether SARS-CoV-2 virus infection impacts renal function differently in patients suffering from ADPKD and CKD when compared to patients suffering only from CKD. Materials and methods: clinical data from 103 patients were collected and retrospectively analyzed. We compared the renal function of ADPKD and CKD patients at two distinct time points: before COVID-19 infection (T0) and 1 year after the infection (T1). We studied also a subpopulation of 37 patients with an estimated glomerular filtration rate (eGFR) < 60 mL/min and affected by ADPKD and CKD. Results: clinical data were obtained from 59 (57.3%) ADPKD patients and 44 (42.7%) CKD patients. At T1, ADPKD patients had significantly higher serum creatinine levels compared to CKD patients, and a significantly lower eGFR was observed only in ADPKD patients with eGFR < 60 mL/min compared to CKD patients (*p* < 0.01, *p* < 0.05; respectively). Following COVID-19 infection, ADPKD–CKD patients exhibited significantly higher variation in both median serum creatinine (*p* < 0.001) and median eGFR (*p* < 0.001) compared to CKD patients. Conclusion: the interplay between COVID-19 and kidney disease is complex. In CKD patients, the relationship between COVID-19 and kidney disease progression is more established, while limited studies exist on the specific impact of COVID-19 on ADPKD patients. Current evidence does not suggest that ADPKD patients are at a higher risk of SARS-CoV-2 infection; however, in our study we showed a significant worsening of the renal function among ADPKD patients, particularly those with an eGFR < 60 mL/min, in comparison to patients with only CKD after a one-year follow-up from COVID-19 infection.

## 1. Introduction

Autosomal Dominant Polycystic Kidney Disease (ADPKD) is one of the most common genetic disorders that affect the kidneys. It is characterized by the development of fluid-filled cysts within the kidneys which can progressively enlarge and impair kidney function over time. Individuals with ADPKD may experience a range of symptoms, including high blood pressure, kidney pain, and, in some cases, kidney failure [1]. The emergence of the COVID-19 pandemic has raised concerns about how this novel coronavirus might impact individuals with preexisting medical conditions, including ADPKD and CKD. COVID-19 is primarily a respiratory illness, but it can have systemic effects on various organs, including the kidneys. Individuals with CKD, such as those with ADPKD, may be at an increased risk of severe complications if they contract the virus [2]. The connection between COVID-19 and ADPKD is not clearly established, although it is known that SARS-CoV-2 virus utilizes Angiotensin-Converting Enzyme 2 (ACE2) receptors on cell membranes to effectively target its host. While it is well-documented that the renin–angiotensin–aldosterone system (RAAS) is more active in ADPKD patients, the extent of hyper-activation or hyper-expression of ACE2 receptors remains unclear. SARS-CoV-2 binding leads to the downregulation of ACE2, potentially disrupting the balance of the RAAS. This can result in increased levels of angiotensin II, which is associated with inflammation, vasoconstriction, and fibrosis. COVID-19 has been linked to acute kidney injury (AKI) and worsening kidney function, particularly in patients with preexisting kidney conditions like ADPKD. The mechanisms include direct viral invasion, cytokine storm, and disruption of the RAAS [3]. Patients with ADPKD are particularly vulnerable due to their already compromised kidney function. The interplay between ADPKD and COVID-19 can lead to a higher risk of AKI and worsening of the renal function. Additionally, the chronic nature of ADPKD means that these patients are at a heightened risk of severe complications from COVID-19, further aggravating their renal condition [4]. Furthermore, the common use of drug treatments involving ACE inhibitors (ACEi) or angiotensin receptor blockers (ARBs) in these patients introduces other clinical implications [1]. Managing COVID-19 in patients with ADPKD requires careful monitoring and treatment adjustments to address both the viral infection and the underlying renal condition.

The aim of the study is to evaluate the progression of renal disease in ADPKD patients as compared to CKD patients after a COVID-19 infection, shedding light on potential differences and implications for clinical management.

## 2. Materials and Methods

### 2.1. Study Design and Subjects

In this study, clinical data from 103 consecutive patients were collected and retrospectively analyzed: 59 patients were affected by ADPKD and 44 patients were affected by CKD. The study aimed to compare the renal function of patients with ADPKD and CKD at two distinct time points: before the COVID-19 infection (T0) and 1 year after the infection (T2). The data were obtained from patients at the University Hospital “Policlinico Umberto I” in Rome. The study protocol was approved by the Clinical Research Ethics Committee at the Sapienza University of Rome, Italy, with Ethics Approval acceptance number 298/2020. The study conforms to the principles outlined in the Declaration of Helsinki and we obtained the written consent from each patient enrolled.

### 2.2. Laboratory Measurements

We performed the following tests with a standard technique: creatinine (mg/dL), serum nitrogen (mg/dL), serum electrolytes (mEq/L), and hemoglobin (g/dL). The eGFR was evaluated according to the Modification of Diet in Renal Disease (MDRD) formula. We evaluated patients with a positive SARS-CoV-2 test using RT-PCR. The negativization of SARS-CoV-2 testing was established following at least two consecutive RT-PCR negative results.

### 2.3. Blood Pressure Measurements

Blood Pressure (BP) measurements were made in the dominant arm after 10 min of rest in the sitting position using a standard automatic sphygmomanometer. The mean of three measurements was recorded. Hypertension was defined according to international guidelines, as SBP ≥ 140 mmHg or DBP ≥ 90 mmHg on repeated measurements.

### 2.4. Statistical Analysis

The coefficient of kurtosis was used to evaluate the normal distribution of data. Continuous variables are expressed as median and interquartile range (IQR) and categorical variables are expressed as absolute frequencies and percentages (%). Group comparisons were made through the Mann–Whitney test or the student’s t test, as appropriate. The chi-square test or Fisher’s exact test, as appropriate, were used to compare categorical variables. A value of *p* < 0.05 was considered to be significant. JASP version 0.17.2.1 software was used for statistical analysis.

## 3. Results

Clinical data from 103 consecutive patients were collected and retrospectively analyzed in this study. A total of 59 (57.3%) patients were affected by ADPKD and 44 (42.7%) patients were affected by CKD.

### 3.1. Demographic and Clinical Characteristics of ADPKD and CKD Patients at T0

Demographic and clinical characteristics of ADPKD and CKD patients at T0 are reported in Table 1. The median age of ADPKD patients was 49 years (IQR 34–57) and 31 (52.5%) patients were female. The median systolic blood pressure was 120 mmHg (IQR 112–130), while the median diastolic blood pressure was 80 mmHg (IQR 75–87) and the median heart rate was 65 bpm (IQR 59–75). Median serum creatinine and blood urea nitrogen levels were estimated at 1.06 mg/dL (IQR 0.88–1.5) and 15.4 mg/dL (IQR 12.7–20.3), respectively, whilst the median eGFR was 65.8 mL/min (IQR 46.8–86.9) and 20 (33.9%) ADPKD patients had an eGFR < 60 mL/min. The median serum albumin was 37 g/L (IQR 34–39) and the median urinary protein level was 12.5 mg/dL (IQR 0–30). ADPKD patients with eGFR < 60 mL/min had significantly higher systolic blood pressure [130 mmHg (IQR 120–137.5) vs. 120 mmHg (IQR 110–130), *p* < 0.05] and diastolic blood pressure [85 mmHg (IQR 80–90) vs. 79 mmHg (IQR 70–86), *p* < 0.05] compared to ADPKD patients with eGFR > 60 mL/min. We did not find any other statistically significant difference between ADPKD patients with normal or reduced eGFR at T0.

The median age of CKD patients was 65.5 years (IQR 56.7–76) and 16 (36.4%) patients were female. The median systolic blood pressure was 140 mmHg (IQR 126–140), the median diastolic blood pressure was 80 mmHg (IQR 70–80), and the median heart rate was 85 bpm (IQR 80–95). Median serum creatinine and blood urea nitrogen levels were 0.9 mg/dL (IQR 0.8–1.35) and 20.5 mg/dL (IQR 12.7–32.2), respectively, whilst the median eGFR was 70 mL/min (IQR 45–87.2) and 17 (38.6%) CKD patients had an eGFR < 60 mL/min. The median serum albumin was 38 g/L (IQR 33.7–38) and the median urinary protein level was 15 mg/dL (IQR 0–30). CKD patients with eGFR < 60 mL/min were significantly older [76 years (IQR 71–77) vs. 60 years (IQR 54–68), *p* < 0.001] compared to CKD patients with eGFR > 60 mL/min. We did not find any other statistically significant difference between CKD patients with normal or reduced eGFR at T0.

### 3.2. Demographic and Clinical Characteristics of ADPKD and CKD Patients at T1

Demographic and clinical characteristics of ADPKD and CKD patients at T1 are reported in Table 2. In ADPKD patients, the median serum creatinine level was 1.1 mg/dL (IQR 0.9–1.7), whilst the median eGFR was 59 mL/min (IQR 41–84.5) and 30 (50.8%) ADPKD patients had an eGFR < 60 mL/min. The median serum albumin level was 36 g/L (IQR 33–39.7) and the median urinary protein level was 20 mg/dL (IQR 10–30). We did not find any other statistically significant difference between ADPKD patients with normal or reduced eGFR at T2. In CKD patients, the median serum creatinine level was 0.89 mg/dL (IQR 0.71–1.32), whilst the median eGFR was 81 mL/min (IQR 50.2–94.2) and 12 (27.3%) CKD patients had an eGFR < 60 mL/min. The median serum albumin level was 33.5 g/L (IQR 30–37) and the median urinary protein level was 15 mg/dL (IQR 0.3–30). We did not find any other statistically significant difference between CKD patients with normal or reduced eGFR at T1. At T1, ADPKD patients with an eGFR < 60 mL/min exhibited a significantly lower eGFR compared to CKD patients (*p* < 0.05).

### 3.3. Comparative Analysis of Renal Function at Each Time Point in ADPKD and CKD Patients

In ADPKD patients, the median serum creatinine was significantly lower at T0 compared to T1 [1.06 mg/dL (IQR 0.88–1.5) vs. 1.1 mg/dL (IQR 0.9–1.7), *p* < 0.001]. In ADPKD patients, the median eGFR was significantly higher at T0 compared to T1 [65.8 mL/min (IQR 46.8–86.9) vs. 59 mL/min (IQR 41–84.5), *p* < 0.05]. There was no statistically significant difference in the variation of eGFR (delta T1_T0) between patients with reduced or normal eGFR at baseline [−6.5 mL/min (IQR −16.2–−1.2) vs. −3.4 mL/min (IQR −15.2–8), *p* > 0.05]. In CKD patients, the median eGFR was significantly lower at T0 compared to T1 [70 mL/min (IQR 45–87.2) vs. 81 mL/min (IQR 50.2–94.2), *p* < 0.05]. There was no statistically significant difference in the variation of eGFR (delta T1_T0) between patients with reduced or normal eGFR at baseline [4 mL/min (IQR 0–19) vs. 9 mL/min (IQR −8.5–16.5), *p* > 0.05]. We observed a significant difference in serum creatinine levels between ADPKD and CKD patients at T1 (and <0.01) (Table 2). Additionally, at T1, ADPKD patients with an eGFR < 60 mL/min exhibited a significantly lower eGFR compared to CKD patients (*p* < 0.05).

### 3.4. Demographic and Clinical Characteristics of a Subpopulation of ADPKD and CKD Patients with eGFR < 60 mL/min

Clinical data from 37 consecutive patients were collected, 20 (54%) patients were affected by ADPKD and CKD and 17 (46%) patients were affected only by CKD. All patients enrolled had an eGFR < 60 mL/min and were affected by systemic arterial hypertension. ADPKD–CKD patients were significantly younger than CKD patients [46 years (IQR 37.5; 63.5) vs. 76 years (IQR 71;77), *p* < 0.001] and female patients were more frequent among ADPKD–CKD patients compared to CKD patients [10 (50%) vs. 3 (17.6%), *p* < 0.05]. Median systolic [130 mmHg (IQR 120;137) vs. 130 mmHg (IQR 120; 142), *p* > 0.05] and diastolic [85 mmHg (IQR 80; 90) vs. 80 mmHg (IQR 73; 81), *p* > 0.05] blood pressure were similar between ADPKD–CKD patients and CKD patients. CKD patients had a significantly higher median heart rate than ADPKD–CKD patients [92 bpm (IQR 80; 101) vs. 56 bpm (IQR 55; 58), *p* < 0.05]. The median C-Reactive Protein (CRP) level was significantly higher in CKD patients compared to ADPKD–CKD patients [110.4 mg/L (IQR 74.9; 156) vs. 1.8 mg/L (IQR 1.1; 2.3), *p* < 0.001], whilst the median serum D-dimer level [1154 mcg/L (IQR 852; 1663) vs. 774 mcg/L (IQR 513; 1304), *p* > 0.05] and the Neutrophil-to-Lymphocyte Ratio (NLR) [6.25 (IQR 5.33; 11.9) vs. 4.2 (IQR 2.5; 7.3), *p* > 0.05] were similar between the cohorts of patients. The median hemoglobin level was similar between ADPKD–CKD patients and CKD patients [11.8 g/dL (IQR 11.4; 12.7) vs. 12.7 g/dL (IQR 11.9; 13.9), *p* > 0.05], whilst the median ratio of arterial oxygen partial pressure (PaO_2_ in mmHg) to fractional inspired oxygen (FiO_2_ expressed as a fraction) (P/F ratio) was significantly lower in CKD patients than in ADPKD–CKD patients [321.5 (IQR 241.7; 358.5) vs. 490 (IQR 471; 504.5), *p* < 0.05]. Table 3 shows demographic and clinical characteristics of the enrolled patients.

At baseline (T0), the median serum creatinine level [1.8 mg/dL (IQR 1.5; 2) vs. 1.7 mg/dL (IQR 1.3; 2.3), *p* > 0.05] and the median eGFR [42.5 mL/min (IQR 39.7; 46.3) vs. 37 mL/min (IQR 27; 54), *p* > 0.05] were similar between ADPKD–CKD patients and CKD patients. Moreover, after SARS-CoV2 infection (T1), the median serum creatinine level [2.2 mg/dL (IQR 1.6; 2.4) vs. 1.5 mg/dL (IQR 1.1; 1.9), *p* > 0.05] and the median eGFR [33.5 mL/min (IQR 26; 44.2) vs. 41 mL/min (IQR 33; 65), *p* > 0.05] were similar between the two cohorts of patients. Table 4 summarizes the comparative analysis of renal function between ADPKD patients and CKD patients. ADPKD–CKD patients had a statistically significant lower median serum creatinine level at T0 than at T1 [1.8 mg/dL (IQR 1.5; 2) vs. 2.2 mg/dL (IQR 1.6; 2.4), *p* > 0.05], with a variation (T1–T0) of 0.29 mg/dL (IQR 0.12; 0.5), and a statistically significant higher eGFR at T0 compared to T1 [42.5 mL/min (IQR 39.7; 46.3) vs. 33.5 mL/min (IQR 26; 44.2), *p* < 0.001], with a variation (T1–T0) of −6.5 mL/min (IQR −16.25; −1.25) (Figure 1A). CKD patients had similar median serum creatinine levels at T0 and at T1 [1.7 mg/dL (IQR 1.3; 2.3) vs. 1.5 mg/dL (IQR 1.1; 1.9), *p* > 0.05], with a variation (T1–T0) of −0.16 mg/dL (IQR −0.3; 0), and a statistically significant lower median eGFR at T0 compared to T1 [37 mL/min (IQR 27; 54) vs. 41 mL/min (IQR 33; 65), *p* < 0.01], with a variation (T1–T0) of 4 mL/min (IQR 0; 19) (Figure 1B).

ADPKD–CKD patients exhibited a significantly larger variation, compared to CKD patients, in both median serum creatinine levels [0.29 mg/dL (IQR 0.12; 0.5) vs. −0.16 mg/dL (IQR −0.3; 0), *p* < 0.001] and median eGFR [−6.5 mL/min (IQR −16.25; −1.25) vs. 4 mL/min (IQR 0; 19), *p* < 0.001] (Table 2 and Figure 1).

## 4. Discussion

The interaction between COVID-19 and kidney disease is intricate, potentially yielding distinct implications for patients with ADPKD and CKD. The study’s longer follow-up period, in comparison to previous reports, offers an opportunity to better understand the impact of SARS-CoV-2 infection on renal function in individuals with ADPKD and CKD. Schmidt-Lauber et al. conducted a comparative analysis, investigating kidney outcomes in 443 patients following mild-moderate COVID-19 alongside a matched control group of 1328 individuals from the general population without prior COVID-19. Their results, assessed over a median follow-up period of 9 months post non-severe COVID-19, revealed only a slight reduction in the mean eGFR. Importantly, no discernible indications pointed toward an elevated risk of progressive kidney dysfunction [5]. Nevertheless, it is noteworthy that approximately 10% of the study cohort had preexisting CKD, and only a subset of these individuals had a specific underlying etiological diagnosis [6]. Similarly, Bowe et al. [7] reported kidney outcomes following COVID-19 in a population-based study involving US veterans. They identified an elevated risk of AKI, eGFR decline, and ESRD after a median follow-up period of around 6 months [8]. Notably, the mean eGFR in this study exceeded 70 mL/min/1.73 m^2^, and the proportion of patients with CKD remained undisclosed [7]. In comparison, our study encompassed patients with different renal diseases, at a higher risk of disease progression. Our cohort exhibited a significantly lower eGFR, and the follow-up period was longer. Consequently, our study could offer an evaluation of whether COVID-19 represents a risk factor for CKD progression in both CKD and ADPKD patients [9]. Indeed, in our cohort, ADPKD patients presented a greater worsening of renal function 1 year after COVID-19 infection; however, this was statistically significant only in patients with eGFR < 60 mL/min. This supports the already known hypothesis that low eGFR can favor the worsening of kidney damage, in our study, in particular, in ADPKD patients. Mirijello et al. [10] showed that patients with reduced eGFR should be considered at high risk of clinical deterioration and death. This further supports the fact that our cohort included patients with a higher risk of CKD progression who may be more prone to worse renal functioning following COVID-19. The impact of SARS-CoV-2 on changes in eGFR has been shown to become more pronounced after a year, suggesting that SARS-CoV-2 infection may have triggered an injury reminiscent of AKI [11]. The association between SARS-CoV-2 infections and AKI is well-established, with proposed mechanisms including virus-related effects as well as those unrelated to the virus (such as sepsis, nephrotoxic agents, cardiovascular instability, etc.) contributing to renal injury. While definitive proof of SARS-CoV-2 renal tropism is lacking, recent data have indicated that SARS-CoV-2-infected pluripotent stem-cell-derived kidney organoids exhibit activation of profibrotic signaling pathways. Hence, SARS-CoV-2 has the capability of directly infecting kidney cells, leading to cellular injury and subsequent fibrosis. These findings offer a potential explanation for both AKI observed in COVID-19 patients and the subsequent development of CKD in individuals experiencing long COVID [11]. Large-scale population studies have sought to evaluate the relationship between CKD and COVID-19 outcomes. A population database study in the United Kingdom, encompassing 17 million patients, revealed a noteworthy association between decreasing eGFR and escalating mortality rates [6]. Individuals with an eGFR ranging from 30 mL/min/1.73 m^2^ to 60 mL/min/1.73 m^2^ faced a 22% heightened risk of death, whereas those with an eGFR lower than 30 mL/min/1.73 m^2^ experienced a 2.5-fold increase in the risk of death. The risk was further elevated in patients on dialysis, the latter experiencing a 3.7-fold increase in the risk of mortality, while those with a solid organ transplant faced a similar 3.5-fold increase in mortality risk [8,9,12]. Specifically focusing on COVID-19 patients with CKD in Mexico, data from 16.000 hospitalized individuals with an eGFR below 60 mL/min/1.73 m^2^ have revealed a striking case fatality rate of 39% [13,14]. Additionally, a meta-analysis involving 16 studies and 871 hospitalized kidney transplant patients with COVID-19 reported a pooled case fatality rate of 24% [15]. Consistent findings emerged from similar retrospective studies conducted in Europe, Turkey, and the United States, highlighting an increased risk of mortality due to COVID-19 associated with decreasing renal function, kidney failure, and transplantation [16,17,18]. While data remain quite limited, there is no evidence to suggest ADPKD increases the likelihood of hospitalization, dialysis, or mortality over and above the risk conferred by the degree of CKD. Other study have not shown an increase in hospitalization, intensive care unit admission, intubation, or mortality following COVID-19 infection in patients with ADPKD compared with patients with other cystic kidney or cystic liver diseases [1,4,5]. While both conditions involve kidney dysfunction, the etiology, progression, and management strategies differ significantly. Limited studies exist on the specific impact of COVID-19 on ADPKD patients. However, it is plausible to speculate that the viral infection may exacerbate the challenges already present in ADPKD. The pro-inflammatory response triggered by COVID-19 could potentially worsen cystic growth, leading to increased pressure within the kidneys and a subsequent decline in renal function. Additionally, the systemic effects of COVID-19, such as hypoxia and coagulopathy, may further compromise kidney health in ADPKD patients [5]. Moreover, it is known that SARS-CoV-2 virus utilizes ACE2 receptors on cell membranes to determine the damage, and it is well-documented that the RAAS is more active in ADPKD patients [1]. The use of ACEIs and ARBs in the context of COVID-19 has been a subject of significant debate and research. ACEIs and ARBs may offer protective effects by mitigating the harmful consequences of elevated angiotensin II levels. By blocking the RAAS pathway, these medications can potentially reduce inflammation and fibrosis, which are exacerbated by COVID-19. There were concerns that ACEIs and ARBs might increase ACE2 expression, potentially facilitating viral entry. However, clinical data have generally not supported this hypothesis. Instead, these medications are thought to stabilize the RAAS, something which might be beneficial in the context of COVID-19. In ADPKD patients, the treatment with ACEi or ARBs is particularly relevant; in fact, these medications are the recommended first-line treatment for hypertension. The HALT-PKD trial has demonstrated that strict blood pressure control with these medications slows the growth of total kidney volume [19]. Early in the pandemic, concerns arose about the use of ACEi and ARBs in COVID-19 patients due to the virus utilizing the ACE2 receptor for cell entry. Competing hypotheses suggest that these medications could either be harmful by increasing the number of ACE2 receptors available for viral binding or beneficial by reducing inflammation and fibrosis leading to lung damage. A systematic review and meta-analysis of 102 observational studies assessed the correlation of ACEi/ARBs use and the likelihood of SARS-CoV-2 infection, mortality, and severe outcomes and showed that prior use of ACEis/ARBs was not associated with an altered risk of SARS-CoV-2 infection or significant change in severe outcomes or mortality after adjustment for confounding factors [20]. The REPLACE COVID trial, a prospective randomized study, was conducted in patients who had previously received a RAAS inhibitor and were admitted to the hospital with COVID-19. A total of 152 participants were enrolled and randomly assigned to either the group continuing or the group discontinuing RAAS inhibitor therapy. The results indicated that participants who continued RAAS inhibition experienced similar lengths of hospital stay, intensive care unit stay, and time on invasive mechanical ventilation compared to those who discontinued RAAS inhibition [21]. Similarly, the BRACE CORONA trial randomized 740 patients with COVID-19 across 29 centers in Brazil to either discontinue or continue ACEi or ARBs. The primary outcome analysis revealed no significant difference in the number of days alive and out of the hospital between the two groups [22]. Studies have shown that continuing ACEIs or ARBs in COVID-19 patients does not increase the risk of severe outcomes and may be associated with improved survival and reduced risk of severe kidney injury. In CKD patients, the relationship between COVID-19 and kidney disease progression is more established. COVID-19 has been associated with AKI, particularly in severe cases [15]. AKI can accelerate the progression of CKD or lead to de novo kidney disease. The virus can directly infect renal cells, causing inflammation and endothelial dysfunction, further contributing to kidney damage. Additionally, the systemic impact of the infection, including cytokine storm and hemodynamic instability, can negatively affect renal function [16]. While both ADPKD and CKD patients face heightened risks following COVID-19 infection, certain nuances differentiate their experiences. ADPKD patients may encounter challenges related to cystic growth and pressure within the kidneys, potentially impacting renal function. In CKD patients, the primary concern lies in the exacerbation of preexisting renal impairment due to COVID-19-induced AKI and systemic effects. Identifying ADPKD patients at higher risk of renal complications following COVID-19 infection can be crucial for early intervention and management. Combining serum and urinary biomarkers of AKI and inflammation and imaging modalities can provide a comprehensive assessment of the risk of renal complications following COVID-19 infection associated with an ADPKD patient. Regular monitoring of serum creatinine, eGFR, and cystatin C can help track kidney function. Urinary biomarkers like NGAL and KIM-1 can provide early warning signs of kidney injury. Imaging studies, particularly MRI or ultrasound, can provide detailed insights into the structural changes in the kidneys [23].

### Limitations of the Study

A limitation of our study is the relatively small cohort of CKD and ADPKD patients. Therefore, additional prospective follow-up studies with a larger number of patients are necessary to confirm our results. Moreover, the follow-up period was relatively short.

## 5. Conclusions

The COVID-19 pandemic has posed challenges for patients with ADPKD and their healthcare providers. Current evidence does not suggest that individuals with ADPKD are at a higher risk of SARS-CoV-2 infection; however, exposure to the virus remains the primary risk factor. For those who contract COVID-19, the risk of hospitalization, intensive care admission, and death is generally increased in individuals with kidney dysfunction, though ADPKD itself does not seem to modify this risk. Despite the lack of increased susceptibility to SARS-CoV-2 infection, our study revealed a significant worsening of renal function among ADPKD patients, particularly those with an eGFR below 60 mL/min, in comparison to patients with CKD after a one-year follow-up from COVID-19 infection. In the nephrological context, it is crucial to stratify patients based on their underlying kidney disease to pinpoint any additional risk or protective factors in the manifestations of COVID-19. This consideration is especially important given the role of the RAAS, a pivotal hormonal system for both kidney function and the evolving pathology of COVID-19. Long-term follow-up is essential to assess the recovery of renal function and mitigate any lasting impacts. Identifying ADPKD patients at higher risk of renal complications following COVID-19 infection involves a multifaceted approach that includes the use of specific biomarkers and imaging modalities. Regular monitoring and early detection are key to managing these risks effectively. Coordination between nephrologists, radiologists, and primary care providers is essential for optimal patient care.

## Figures and Tables

**Figure 1 biomedicines-12-01301-f001:**
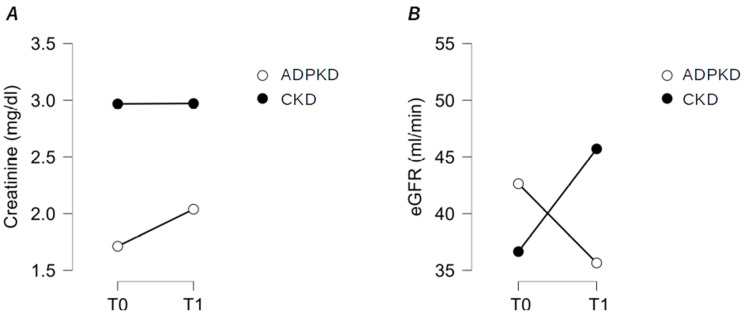
Comparative analysis between 20 ADPKD patients and 17 CKD patients. (**A**) Variation in serum creatinine levels from baseline (T0) to T1 (after SARS-CoV-2 infection); (**B**) variation in serum estimated Glomerular Filtration Rate (eGFR) from baseline (T0) to T1 (after SARS-CoV-2 infection).

**Table 1 biomedicines-12-01301-t001:** Demographic and clinical characteristics of ADPKD and CKD patients at T0. All data are expressed as median and interquartile range (IQR) or as absolute frequency and percentage (%).

	ADPKD (*n* = 59)	CKD (*n* = 44)	*p*
Age, years	49 (34–57)	65.5 (56.7–76)	<0.001
M/F	28 (47.5)/31 (52.5)	28 (63.6)/16 (36.4)	>0.05
Systolic blood pressure, mmHg	120 (112–130)	140 (126–140)	<0.001
Diastolic blood pressure, mmHg	80 (75–87)	80 (70–80)	>0.05
Heart rate, bpm	65 (59–75)	85.5 (80–97.2)	<0.001
SpO_2_, %	98.9 (97.5–99.4)	95 (94–97)	<0.001
Hb, g/dL	12.6 (12–13.8)	13.1 (12–14)	>0.05
Serum creatinine, mg/dL	1.06 (0.88–1.5)	0.9 (0.8–1.35)	>0.05
eGFR, mL/min	65.8 (46.8–86.9)	70 (45–87.2)	>0.05
eGFR < 60 mL/min	20 (33.9)	17 (38.6)	>0.05
Blood urea nitrogen, mg/dL	15.4 (12.7–30.3)	20.5 (12.7–35.2)	<0.001
Na^+^, mEq/L	141 (139–144)	135 (132–138)	<0.001
K^+^, mEq/L	4.1 (4–4.4)	3.9 (3.6–4.4)	>0.05
Ca^2+^, mg/dL	9.4 (9–10)	8.5 (8.3–9.2)	>0.05
Serum albumin, g/L	37 (34–39)	38 (33.7–41)	>0.05
Urinary proteins, mg/dL	12.5 (0–30)	15 (0–30)	>0.05

**Table 2 biomedicines-12-01301-t002:** Demographic and clinical characteristics of ADPKD and CKD patients at T1. All data are expressed as median and interquartile range (IQR) or as absolute frequency and percentage (%).

	ADPKD (*n* = 59)	CKD (*n* = 44)	*p*
Serum creatinine, mg/dL	1.1 (0.9–1.74)	0.89 (0.71–1.32)	<0.01
eGFR, mL/min	59 (41–84.5)	81 (50.2–94.2)	>0.05
eGFR < 60 mL/min	28 (47.4)	12 (27.3)	<0.05
Serum albumin, g/L	36 (33–39.7)	33.5 (30–37)	>0.05
Urinary proteins, mg/dL	20 (10–30)	15 (0.3–30)	>0.05

**Table 3 biomedicines-12-01301-t003:** Demographic and clinical characteristics of ADPKD and CKD patients. All data are expressed as median and interquartile range (IQR) or as absolute frequency and percentage (%).

	ADPKD (*n* = 20)	CKD (*n* = 17)	*p*
Age, years, median (IQR)	46 (37.5–63.5)	76 (71–77)	<0.001
M/F, *n* (%)	10 (50)/10 (50)	3 (17.6)/14 (82.4)	<0.05
SBP, mmHg, median (IQR)	130 (120–137)	130 (120–142)	>0.05
DBP, mmHg, median (IQR)	85 (80–90)	80 (73–81)	>0.05
Heart rate, bpm, median (IQR)	56 (55–58)	92 (80–101)	<0.05
CRP, mg/L, median (IQR)	1.8 (1.1–2.3)	110.4 (74.9–156)	<0.001
D-dimer, mcg/L, median (IQR)	774 (513–1304)	1154 (852–1663)	>0.05
NLR, median (IQR)	4.2 (2.5–7.3)	6.25 (5.33–11.9)	<0.001
Hb, g/dL, median (IQR)	11.8 (11.4–12.7)	12.7 (11.9–13.9)	>0.05
P/F ratio, median (IQR)	490 (471–504.5)	321.5 (241.7–358.5)	<0.05

ADPKD: Autosomal Dominant Polycystic Kidney Disease; CKD: Chronic Kidney Disease; M: male; F: female; SBP: Systolic Blood Pressure; DBP: Diastolic Blood Pressure; CRP: C-Reactive Protein; NLR: Neutrophil-to-Lymphocyte Ratio; Hb: hemoglobin.

**Table 4 biomedicines-12-01301-t004:** Comparative analysis of renal function between 20 ADPKD patients and 17 CKD patients at baseline T0 and after SARS-CoV2 infection (T1). All results are expressed as median and interquartile range (IQR).

	T0			T1		
	ADPKD	CKD	*p*	ADPKD	CKD	*p*
Creatinine, mg/dL	1.8 (1.5; 2)	1.7 (1.3; 2.3)	>0.05	2.2 (1.6; 2.4)	1.5 (1.1; 1.9)	>0.05
eGFR, mL/min	42.5 (39.7; 46.3)	37 (27; 54)	>0.05	33.5 (26; 44.2)	41 (33; 65)	>0.05
	Variation T1–T0					
	ADPKD			CKD		*p*
Creatinine, mg/dL	0.29 (0.12; 0.5)			−0.16 (−0.3; 0)		<0.001
eGFR, mL/min	−6.5 (−16.25; −1.25)			4 (0; 19)		<0.001

ADPKD: Autosomal Dominant Polycystic Kidney Disease; CKD: Chronic Kidney Disease; eGFR: estimated Glomerular Filtration Rate.

## Data Availability

Data are available upon request.

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
