# Peer review of "COVID-19 Infection in Autosomal Dominant Polycystic Kidney Disease and Chronic Kidney Disease Patients: Progression of Kidney Disease"

_biomedicines, 2024, doi:10.3390/biomedicines12061301_

Round 1

Reviewer 1 Report

Comments and Suggestions for Authors

Progression of Kidney Disease in Autosomal Dominant Polycystic Kidney Disease and Chronic Kidney Disease Patients Following COVID-19 Infection

The title should be modified to make it clearer and catchier to the reader.

The manuscript is well written, and it is of clinical interest. I have the following minor comments for authors to consider: 

The conclusion in the abstract should be rewritten, and it should imply only the inferences of the study results.

The authors should consider reducing the length of the abstract.

The introduction should be expanded by citing all the pertinent literature. The rationale for this should be highlighted.

Was the sample size required for the study calculated? If yes, have authors reached a sufficient sample?

The results and discussion are well written and clear.

Comments on the Quality of English Language

Minor editing of English language required.

Author Response

Rebuttal Letter

Dear Editor,

Thank you very much for giving us the chance to revise our original article (biomedicines-2999785, entitled "Progression of Kidney Disease in autosomal dominant polycystic kidney disease and chronic kidney disease patients following COVID-19 Infection"). We are grateful to editor and reviewers for their comments and suggestions. The point-by-point responses to the comments from the editor and the reviewers have been addressed. The changes in the text are in red color.

We hope that the manuscript is now suitable for publication in Biomedicines.

Looking forward to hearing from you soon.

Yours sincerely

Silvia Lai,

Associate Professor

Department of Translational and Precision Medicine,

Sapienza University of Rome, Italy

Viale del Policlinico 155, 00161 Rome.

Reviewers'comments:

Reviewer #1

Comments and Suggestions for Authors

Progression of Kidney Disease in Autosomal Dominant Polycystic Kidney Disease and Chronic Kidney Disease Patients Following COVID-19 Infection

The title should be modified to make it clearer and catchier to the reader.

Authors’reply: we thanks the Reviewer for the interesting suggestions. We changed the title of the manuscript

COVID 19 infection in autosomal dominant polycystic kidney disease and chronic kidney disease patients: progression of Kidney Disease

The manuscript is well written, and it is of clinical interest. I have the following minor comments for authors to consider:

Authors’reply: we thank the Reviewer very much for his words

The conclusion in the abstract should be rewritten, and it should imply only the inferences of the study results.

Authors’reply: we thanks the Reviewer for suggestion. We modified the conclusion of the abstract.

The authors should consider reducing the length of the abstract.

Authors’reply: we thanks the Reviewer, we reduced the length of the abstract.

The introduction should be expanded by citing all the pertinent literature. The rationale for this should be highlighted.

Authors’reply: we thanks the Reviewer for the interesting suggestions. We modified the text

Was the sample size required for the study calculated? If yes, have authors reached a sufficient sample?

Authors’reply: I agree with the reviewer, however methodologies such as retrospective study may be utilized in settings for which existing datasets are not available. While specifying the number of sample to be extracted and/or determining whether the number that can feasibly extracted will be clinically meaningful is an important study design consideration, there is a lack of rigorous methods available for sample size calculation in this setting. So it is difficult to evaluate if the sample size is adequate.

The results and discussion are well written and clear.

Authors’reply: we thank the Reviewer very much for these words

Comments on the Quality of English Language

Minor editing of English language required.

Authors’reply: we thanks the Reviewer, native english speaker revised the manuscript

Reviewer 2 Report

Comments and Suggestions for Authors

Lai et al. submitted the article titled: "Progression of Kidney Disease in autosomal dominant polycystic kidney disease and chronic kidney disease patients following COVID-19 Infection".

The manuscript is nicely written, clear and understandable. Minor typos are detected (please go through the text and try to use uniform type of writing (e.g. COVID-19).

The Introduction could be expanded to give more significant inputs related to CKD and ADPKD and their connection to COVID-19.

Future Research Directions are missing. It would be valuable if the authors discussed potential avenues for future research in this area. For instance, are there specific biomarkers or imaging modalities that could help identify ADPKD patients at higher risk for renal complications following COVID-19 infection? Are there any therapeutic interventions that could mitigate the impact of COVID-19 on renal function in this population?

  Comments on the Quality of English Language

Author Response

Rebuttal Letter

Dear Editor,

Thank you very much for giving us the chance to revise our original article (biomedicines-2999785, entitled "Progression of Kidney Disease in autosomal dominant polycystic kidney disease and chronic kidney disease patients following COVID-19 Infection"). We are grateful to editor and reviewers for their comments and suggestions. The point-by-point responses to the comments from the editor and the reviewers have been addressed. The changes in the text are in red color.

We hope that the manuscript is now suitable for publication in Biomedicines.

Looking forward to hearing from you soon.

Yours sincerely

Silvia Lai,

Associate Professor

Department of Translational and Precision Medicine,

Sapienza University of Rome, Italy

Viale del Policlinico 155, 00161 Rome.

Reviewers'comments:

Reviewer #2

Comments and Suggestions for Authors

Lai et al. submitted the article titled: "Progression of Kidney Disease in autosomal dominant polycystic kidney disease and chronic kidney disease patients following COVID-19 Infection".

The manuscript is nicely written, clear and understandable.

Minor typos are detected (please go through the text and try to use uniform type of writing (e.g. COVID-19).

Authors’reply: we thanks the Reviewer for this observation. We modified the text.

The Introduction could be expanded to give more significant inputs related to CKD and ADPKD and their connection to COVID-19.

Authors’reply: we thanks the Reviewer for the interesting suggestions. We modified the text.

Future Research Directions are missing. It would be valuable if the authors discussed potential avenues for future research in this area. For instance, are there specific biomarkers or imaging modalities that could help identify ADPKD patients at higher risk for renal complications following COVID-19 infection? Are there any therapeutic interventions that could mitigate the impact of COVID-19 on renal function in this population?

Authors’reply: we thanks the Reviewer for the interesting suggestions. We modified the text.

Reviewer 3 Report

Comments and Suggestions for Authors

Dear Authors, I was reviewing with interest the manscript entiteled "Progression of Kidney Disease in autosomal dominant polycystic kidney disease and chronic kidney disease patients following COVID-19 Infection". The paper deals with a very interesting topic, especially the ADPKD-patients and their outcome in the context of COVID-19. It's a pity, that only a small number of patients have been investigated, so the conclusions remain speculative. However, the data on ADPKD-patients are novel and interesting. In the introduction, the potential relation between patient medication (ACE-inhitors/ARBs) and impact of COVID-19 on kidney function has been mentioned. It might improve the value of the manuscript, to show data on this topic of the investigated patients. 

Author Response

Rebuttal Letter

Dear Editor,

Thank you very much for giving us the chance to revise our original article (biomedicines-2999785, entitled "Progression of Kidney Disease in autosomal dominant polycystic kidney disease and chronic kidney disease patients following COVID-19 Infection"). We are grateful to editor and reviewers for their comments and suggestions. The point-by-point responses to the comments from the editor and the reviewers have been addressed. The changes in the text are in red color.

We hope that the manuscript is now suitable for publication in Biomedicines.

Looking forward to hearing from you soon.

Yours sincerely

Silvia Lai,

Associate Professor

Department of Translational and Precision Medicine,

Sapienza University of Rome, Italy

Viale del Policlinico 155, 00161 Rome.

Reviewers'comments:

Reviewer #3

Comments and Suggestions for Authors

Dear Authors, I was reviewing with interest the manscript entiteled "Progression of Kidney Disease in autosomal dominant polycystic kidney disease and chronic kidney disease patients following COVID-19 Infection". The paper deals with a very interesting topic, especially the ADPKD-patients and their outcome in the context of COVID-19. It's a pity, that only a small number of patients have been investigated, so the conclusions remain speculative. However, the data on ADPKD-patients are novel and interesting.

Authors’reply: we thanks the Reviewer

In the introduction, the potential relation between patient medication (ACE-inhitors/ARBs) and impact of COVID-19 on kidney function has been mentioned. It might improve the value of the manuscript, to show data on this topic of the investigated patients.

Authors’reply:  we thanks the Reviewer for the interesting suggestions. We added it in the text

Round 2

Reviewer 3 Report

Comments and Suggestions for Authors

.